# The Effect of Socioeconomically Disadvantaged Stereotype Threat on Inhibitory Control in Individuals with Different Household Incomes

**DOI:** 10.3390/bs13121016

**Published:** 2023-12-18

**Authors:** Shanshan Wang, Dong Yang

**Affiliations:** 1School of Psychology, Hainan Normal University, Haikou 571158, China; 2Department of Psychology, Southwest University, Chongqing 400700, China

**Keywords:** socioeconomically disadvantaged stereotype threat, inhibitory control, monetary effect, household income

## Abstract

Previous studies have discussed the impact of the socioeconomically disadvantaged stereotype threat (SDST) on inhibitory control. But the specific influences of the SDST on inhibitory control in different household income groups are not clear. We hypothesized that the SDST had different effects on inhibitory control in individuals with distinct household income, and the attribution of stimuli would influence it as well, especially the currency value of the stimuli. To investigate it, two studies were conducted, which required inhibiting their motor responses. Specifically, Study 1 explored the influence of the SDST on basic inhibitory control. Study 2 analyzed the influence of the SDST on inhibitory control when the input stimuli included currency values and monetary conception. The results revealed that the inhibitory control ability was worse in the lower income group but not during the processing of stimuli with currency value. For the effect of the SDST, it found that there was a negative effect on those with a lower household income and a positive effect on those with a higher household income. Based on the findings, the effect of the SDST on inhibitory control in human beings is not stable; instead, it varies depending on the traits of the stimuli in different tasks and of the individuals themselves.

## 1. Introduction 

Poverty, defined as socioeconomically disadvantaged status, has long-term effects on working memory [1], executive function [2,3], and attention [4,5]. Moreover, the effects of being socioeconomical disadvantaged are impact all age groups: researchers have found that being socioeconomical disadvantaged is associated with impaired cognitive abilities in infants [6], children, adolescents [7,8,9], and adults [10,11]. For instance, it found that the cerebral areas of language, reading, executive control, memory, and other cognitive abilities of children growing up in poor families were smaller in size than those who grew up in affluent families [12]. 

Over time, stereotypes of the poor have formed. However, out of a fear that the negative stereotypes will be confirmed, the behavior may become worse [13,14,15,16]. This is the typical effect of a stereotype threat. Therefore, the socioeconomically disadvantaged stereotype threat (SDST) results from the fact that, due to the negative stereotypes of persons with socioeconomically disadvantaged status, when those persons belong to socioeconomically disadvantaged groups, the behavior becomes worse due to a fear that the negative stereotypes will be confirmed. For this, Schmader and his colleagues (2008) raised an integrated process model regarding stereotype threat effects on performance. In this model, it showed that a stereotype threat had effects on physiological stress response, monitoring processes, working memory, cognitive control, and so on [14]. A common stereotype associated with socioeconomical disadvantage is that the intelligence of the poor is lower than of the rich. It found that people from lower income families who experienced a stereotype threat performed worse on intelligence tests than those from higher income families [17]. Taking the intelligence test, the lower income participants who were informed performed worse than those who were not informed [18,19]. These findings verify that income-based stereotypes threat have negative effects on the lower income ones. 

However, in reality, stereotype threat can sometimes result in improved performance. For example, when marginalized individuals experience stereotype threat, due to a reluctance to affirm the negative stereotypes belonging to their group, they perform better to confute the negative stereotypes [16,20,21]. In other words, stereotype threat have both negative and positive effects on those who experience them. 

Thus, we assumed that the effect of the SDST is not only negative but could be positive as well. 

Furthermore, the theory of poverty’s influence on cognitive control showed that having the status of being socioeconomically disadvantaged can lead to a depletion of cognitive resources and a loss of cognitive control [22,23]. As we know, inhibitory control is the ability to monitor and address conflict in advanced cognitive processing, which is closely related to decision-making, inhibition, working memory, cognitive flexibility, and impulse control [24,25,26,27], all of which monitor and regulate cognitive functions. Whether the effect of the SDST on the inhibitory control comprehensively and whether the effect of the SDST is different or the same in people with different household income levels are not clear. Furthermore, it is worth nothing that the psychological activities associated with stereotypes are not entirely automatic, instead being controlled or inhibited consciously [28]. External elements, such as input stimuli, social comparison, and so on, are required to induce effects of negative stereotypes, and the effects of stereotype threat will vary depending on the attributes of the group or the surroundings [29,30,31], such as the characteristics of input stimuli and the identity of the negative stereotypes in the ingroup. For example, anxious people are highly responsive to negative stimuli related to anxiety [32], while those with depression are responsive to negative information and are less likely to respond to positive information [33]. In addition, individuals are more focused on gender stereotypes that are consistent with their own gender, rather than the stereotypes associated with other genders [34]. Thus, when a stereotype threat is present, there are varying levels of responsiveness to different input stimuli, and this can lead to differences in behaviors. Therefore, theoretically, stereotype threat of people who are socioeconomical disadvantaged should have effects on the inhibitory control ability as well, which is not very clear. To verify it, this research aimed to explore the effects of the SDST on the inhibitory control capacity, which is an essential factor of cognitive processing. 

### 1.1. Hypothesis

Based on the findings above, there were some hypotheses that were assumed before conducting the experiments, which the findings of the study sought to prove. 

**H1.** 
*The SDST has an effect on the inhibitory control ability, and these effects differ based on household income level.*


Some findings can promote this hypothesis. For example, opposing effects of the monetary conception on individuals have been found [35,36]. The slight mention of the monetary conception can cause changes in behaviors, such as reducing helping behavior, decreasing intimate physical behaviors, exhibiting a preference toward independent activities, and working harder [37]. This shows several negative effects, but also a positive effect. Regarding the experiences of socioeconomical disadvantage, it has been found that short-term socioeconomically disadvantaged experiences are perhaps beneficial for cognitive ability, but long-term socioeconomically disadvantaged experiences may be harmful [38]. Socioeconomical disadvantage is accompanied by a state of scarcity, such as a lack of financial or material resources. In an emergency, scarcity forces individuals to focus on the rational utilization of limited resources [39]; however, scarcity itself also utilizes attention resources and reduces cognitive ability and executive control capabilities [5]. These findings are contrary to the suggestion that being socioeconomical disadvantaged not only has a negative impact on an individual’s performance, but has few positive effects, such as working hard. Thus, we believe that the relative effects of the SDST are both both negative and positive, though they depend on the individuals themselves. 

**H2.** 
*An awareness of socioeconomical disadvantage or socioeconomical advantage (poverty or wealth) depending on the input stimuli influences the effect of the SDST on the inhibitory control ability.*


As an environmental factor, people can become conscious of socioeconomical disadvantage or socioeconomical advantage by making upper or downward social comparisons. Gopinath and Nair found that awareness of one’s own economic situation has a more profound impact on the poor [40]. Haushofer and Fehr found that income shocks—that is, sudden and unexpected drops in income—have more significant impacts on those who are poor than those who are not [41]. Therefore, awareness of one’s own economic situation may also have different effects on different groups. Then, the characteristics of the stimuli could arouse the awareness of socioeconomical disadvantage or socioeconomical advantage because there were some stimuli describing the status of socioeconomical disadvantage and some of socioeconomical advantage. Take, for example, the images of the houses of the where the poor and the rich live. However, stereotype threat cause people to pay attention to negative information [13] and utilize limited cognitive resources. This is an instinctive reaction that allows individuals to confirm the source of the threatening information [42]. However, it also causes distractions, prompting people to focus on irrelevant information that is not conducive to completing the tasks at hand. As the result, overall performance outcomes are reduced [13,43]. Thus, as the relative and affected factor of socioeconomical disadvantage, the awareness of socioeconomical disadvantage or socioeconomical advantage should be aroused by the processing of input stimuli. This awareness may also have an interaction effect with the SDST on an individual’s inhibitory control, which plays an essential role in cognitive performance.

To explore the assumed hypothesis, two studies were conducted. Study 1 explored the impact of the SDST on the basic inhibitory control ability in different income groups. The typical stop-signal task was conducted. This task required the suppression and execution with distinct reactions for smaller probability events. The stimuli were neutral and were not associated with the conceptions of socioeconomical disadvantage. 

Regarding the influences of the input stimuli, Study 2 was conducted to explore the interaction effects of stimuli on the impact of the SDST on the inhibitory control. Based on the conceptions of socioeconomical disadvantage, the stimuli with currency value and the labels of socioeconomical disadvantage and socioeconomical advantage were settled in the studies, respectively. 

Based on the scarcity mentality of socioeconomical disadvantage, Study 2 was conducted with the stimuli, with some of them having currency value. It mainly explored the effects of the SDST on the inhibitory control, which required the suppression of the stimuli with currency value. Additionally, it explored the influences of socioeconomically advantaged awareness during the task. In sum, the main aim of this research was to explore the effects of the SDST on the inhibitory control capacity in individuals with distinct household income levels. 

### 1.2. Sample Size Consideration

Sample size was determined a priori by utilizing G*Power 3.1.9.4 [44] for repeated analysis of variance (ANOVA). As for the action stimulation paradigms [45], the large effect size was set to the parameters as follows: effect size f = 0.25, alpha level α = 0.05, and power (1-β err prob) = 0.95. The calculation suggested a minimum total sample size of 76 for Study 1 (repeated-measures ANOVA for 2 (Group: LG vs. HG) × 2 (Threat condition: threat vs. non-threat) × 2 (Directional judgment: same arrow vs. different arrow)), 52 for Study 2 (repeated-measures ANOVA for 2 (Group: LG vs. HG) × 2 (Threat condition: threat vs. non-threat) × 2 (Part: 1st vs. 2nd) × 2 (Items: GI vs. NGI or IGI vs. NGI or FGI vs. IGI), in which the Group and Threat conditions were between-subject factors and others were within-subject factors).

## 2. Method

The operation steps regarding threat conditions were each study initially. Specifically, first the demographic data were collected, including gross household income, area of the long-term dwelling, number of family members, income sources, family members who could not make money, and so on. Then, the participants read a brief essay of about 450–500 words based on the research paradigm of stereotype threat. All reading materials were written in the form of scientific reports and newspapers. There were two essays. One essay depicted images taken by an asteroid detector and was assigned to the non-threatened group. Another essay discussed the scientific findings of the impact of poverty on cognitive abilities. The group that was assigned to read this essay was defined as the threatened group.

To check the threatened condition, two open questionnaire surveys were conducted. The first survey was a self-assessment before the studies. Another survey was completed after finishing the studies, and it dealt with the stereotypes of socioeconomically disadvantaged people. The frequencies of the words from the answers were analyzed. We found that the frequencies of poverty-related (poor, poverty), inferiority-related (inferiority), and anxiety-related words were significantly higher in the threatened group than in the non-threatened group. The results helped to verify that the socioeconomically disadvantaged stereotype threat was successfully aroused. Specifically, the frequencies of poverty-related, inferiority-related, and anxiety-related words (inferiority-related, *p* < 0.001; poverty-related, *p* = 0.006; anxiety-related, *p* = 0.002) in the first survey were significantly greater in the threatened group than in the non-threatened group. In the second survey, the frequency of inferiority-related words was significantly higher than that of other types of words in the threatened group (*p* < 0.001). 

## 3. Study 1 

### 3.1. Aim

Based on H1, Study 1 investigated the effect of the SDST on the basic inhibitory control ability in the lower household income group and the higher household income group. 

### 3.2. Participants 

One hundred and nine university students participated in this study (average age = 20.39). Based on the household income level, the groups were classified. Rural residents and urban residents with a per capita income of less than 1000 RMB and 2000 RMB, respectively, were sorted into the lower household income group. Fifty-three participants were from families with the lower household income group (LG). The remaining participants belonged to the higher household income group (HG; N = 56). Furthermore, the participants were assigned to the threatened condition randomly by the traditionally threatened method. The survey results were checked in order to confirm whether the participants were under the threatened condition. Therefore, 28 LG participants and 28 HG participants were under the SDST, and the rest of the participants were not (LG N = 25, HG N = 28). 

### 3.3. Procedure and Data Collection

Each trial began with a white fixation on a black screen for 500–800 ms. Next, a white or red circle appeared randomly for 400 ms. After a 400 ms blank screen, a left or right green arrow was displayed on the screen for 1000 ms. If the white circle had preceded the arrow, the participants were required to press a key corresponding to the direction of the arrow. If the red circle had preceded the arrow, the participants were required to press a key corresponding to the opposite direction of the arrow. For example, after the white circle, if there was a left arrow, the correct response was to press the left arrow key. If there had been a red circle, the correct response was to press the right arrow key. In this task, there were a total of 240 trials with the white circle and 80 with the red circle (see Figure 1). This task required participants to respond as quickly as possible. Reaction time (RT) and accuracy (ACC) were recorded. 

### 3.4. Data Analysis 

Repeated ANOVA were performed regarding the RT and ACC data, which used the following design: 2 (Group: LG vs. HG) × 2 (Threat condition: threat vs. non-threat) × 2 (Directional judgment: same direction arrow vs. different direction arrow). A simple effect analysis was then performed to determine the interaction effects with significant differences. The results were adjusted using sphericity and the Greenhouse–Geisser correction. A post hoc comparison and the Bonferroni correction were applied to the simple effect analysis using SPSS (version 22.0).

### 3.5. Results

The ACC results showed significant interaction effects in the following: 2 (Group: LG vs. HG) × 2 (Threat condition: threat vs. non-threat) × 2 (Directional judgment: same vs. different) (F(1,105) = 4.89, *p* = 0.03, ŋ_p_^2^ = 0.04) and 2 (Group: LG vs. HG) × 2 (Threat condition: threat vs. non-threat) (F(1,105) = 4.26, *p* = 0.04, ŋ_p_^2^ = 0.04).

Further analysis found that participants in LG exhibited a less directional judgment ACC of the different arrow than that of the same arrow, regardless of whether they were experiencing a threat or not (threat: F(1,105) = 8.13, *p* = 0.01, ŋ_p_^2^ = 0.07, Cis = [0.01,0.05]; non-threat: F(1,105) = 31.70, *p* < 0.001, ŋ_p_^2^ = 0.23, Cis = [0.04,0.08]). When HG was experiencing a threat, the directional judgment ACC of the different arrow was significantly less than that of the same arrow (threat: F(1,105) = 9.97, *p* = 0.01, ŋ_p_^2^ = 0.09, CIs = [0.01,0.05]; non-threat: F(1,105) = 3.16, *p* = 0.08, ŋ_p_^2^ = 0.03, CIs = [−0.01,0.04]) (see Figure 2). 

For the different directional judgment, when not under threat, the ACC of HG was significantly greater than that of LG (LG: M ± SE = 0.89 ± 0.02, CIs = [0.86,0.92], HG: M ± SE = 0.95 ± 0.01, CIs = [0.92,0.98], F(1,105) = 7.52, *p* = 0.01, ŋ_p_^2^ = 0.67). In LG, for the Different judgment, the ACC of the threatened condition was significantly greater than of the non-threatened condition (threat: M ± SE = 0.94 ± 0.01, CIs = [0.91,0.97], non-threat: M ± SE = 0.89 ± 0.02, CIs = [0.86,0.92], F(1,105) = 5.38, *p* = 0.02, ŋ_p_^2^ = 0.05).

When under the non-threatened condition, the ACC of HG was significantly greater than that of LG (LG: M ± SE = 0.92 ± 0.01, CIs = [0.89,0.94], HG: M ± SE = 0.96 ± 0.01, CIs = [0.93,0.98], F(1,105) = 4.37, *p* = 0.04, ŋ_p_^2^ =0.40). There were no significant differences in the threatened condition (LG: M ± SE = 0.95 ± 0.01, CIs = [0.93,0.98], HG: M ± SE = 0.94 ± 0.01, CIs = [0.91,0.96], F(1,105) =0.66, *p* = 0.42, ŋ_p_^2^ = 0.01).

There were no significant differences in RT. Specifically, the interaction effect of Group × Threat condition × Directional judgment was not significant (F(1,105) = 0.05, *p* = 0.83, ŋ_p_^2^ < 0.001), as well as the interaction effect of Group × Threat condition (F(1,105) = 0.55, *p* = 0.46, ŋ_p_^2^ = 0.01) (see Figure 3). 

### 3.6. Discussion

In this task, when the red circle appeared, participants were required to select the key corresponding to the opposite direction of the arrow. The participants were required to inhibit the impulsive reaction of selecting the same arrow (as they would when the white arrow appeared) and perform a different behavior from these items with smaller ratio. This experiment allowed us to study the basic inhibitory control. 

First, the results showed that the LG group exhibited worse inhibitory control as compared to the situation in which the white circle appeared, when no inhibitory control was required. This reflects the weak inhibitory control ability of LG. 

Second, the results indicated that, when individuals were not under threat, all performance was significantly better in HG than in LG. This reveals that HG have better inhibitory control overall, which reflects the differences in behaviors of each group. However, when the SDST was applied, the group differences disappeared. This shows that the SDST decreases the group differences in this study. Specifically, it narrows the differences in performance of inhibitory control. 

Further analysis found that the SDST allowed LG to perform better with higher scores but that it hindered HG with worse performance outcomes of inhibitory control. These results verify the existence of the effect of the SDST, as well as the positive effect that it has on LG and the negative effect it has on HG. This shows that the SDST has opposing effects on these groups, and this could negate the original group differences in inhibitory control.

In sum, the results revealed that (1) LG has weak inhibitory control ability overall; (2) LG has worse inhibitory control ability than HG; and (3) LG and HG experience opposing effects from the SDST on inhibitory control, which eliminates the original group differences. 

## 4. Study 2

### 4.1. Aims

This study explored the effect of the SDST on the inhibitory control of different groups when the input stimuli are related to monetary conception (currency value). It also investigated the effect of aroused socioeconomically disadvantaged/advantaged awareness in this study.

### 4.2. Participants

The criteria for the participants were the same as in Study 1. One hundred and ten university students participated in the experiment (average age = 20.40). Fifty-four participants were sorted into the LG group. The rest of the participants were sorted into the HG group (N = 56). Twenty-eight LG participants and 28 HG participants were under the SDST, while the remainder were not (LG N = 26, HG N = 28). 

### 4.3. Procedure and Data Collection

The materials in this experiment were the round gold coin, the round silver coin, and the solid circle in pure color. The colors of the solid circle included four colors: green, yellow, and the colors of the gold coin and the silver coin. 

In this experiment, no responses for the gold coin were required; these were referred to as the NoGo items (NGI). The rest of the stimuli required the same button response, so these were called the Go items (GI). Participants were asked to respond as quickly and correctly as possible in the limited time. Each trial began with a white fixation on a black screen for 800 ms. The stimuli then appeared on the screen for 400 ms. The participants were required to respond in 1000 ms. In the experiment, there were 40 trials with the gold coin (NGI), 40 trials with the silver coin (Infrequency Go Items; IGI), and 240 trials with the pure color solid circle (Frequency Go Items; FGI). Therefore, the ratio of NGI to IGI to FGI was 1:1:6, and the ratio of Go to NoGo items was 7:1. The response for the silver coin was the small probability event in this study (see Figure 4B).

This experiment was divided into two parts. At the beginning, participants were informed that their performance outcomes would automatically be converted into the accumulation of wealth by the computer. Performances with higher ACC and shorter RT accumulated a higher wealth value (between 0.1 and 1 RMB) of each trial. After finishing the Pre-task, participants were provided with feedback of their wealth accumulation to let the participants know about their performances and to arouse the awareness of socioeconomical disadvantage and socioeconomical advantage (see Figure 4A). Several minutes later, Pre-task was repeated in its entirety as Post-task, and participants were informed that the accumulated wealth would be refreshed. 

In this experiment, the feedback was illusory and had two types. The feedback for the threatened participants showed that their accumulation of wealth was significantly less than others to arouse the socioeconomically disadvantaged awareness of their own. And the feedback for the non-threatened participants showed that their accumulation of wealth was significantly more than others, in order to arouse the socioeconomically advantaged awareness of their own. 

### 4.4. Data Analysis 

To verify the effect of the SDST, the effect of the aroused socioeconomically advantaged awareness, and their interaction effect, the repeated ANOVA was performed as follows:(1)To determine inhibitory control, the following repeated ANOVA was conducted: 2 (Group: LG vs. HG) × 2 (Threat condition: threat vs. non-threat) × 2 (Task: Pre- vs. Post-) × 2 (Items: GI vs. NGI).(2)For inhibitory control based on monetary value (or monetary conception), the following repeated ANOVA was conducted: 2 (Group: LG vs. HG) × 2 (Threat condition: threat vs. non-threat) × 2 (Task: Pre- vs. Post-) × 2 (Items: IGI vs. NGI) and 2 (Group: LG vs. HG) × 2 (Threat condition: threat vs. non-threat) × 2 (Task: Pre- vs. Post-) × 2 (Items: FGI vs. IGI).

Following this, a simple effect analysis was carried out. All results were adjusted using sphericity and the Greenhouse–Geisser correction. A post hoc comparison and the Bonferroni correction were applied to the simple effect analysis using SPSS (version 22.0).

## 5. Results 

### 5.1. Differences in Inhibitory Control Regarding the Monetary Value Effect

In this experiment, the differences of NGI and GI reflect inhibitory control. Thus, the repeated ANOVA—2 (Group: LG vs. HG) × 2 (Threat condition: threat vs. non-threat) × 2 (Task: Pre- vs. Post-) × 2 (Items: GI vs. NGI)—was performed. Significant interaction effects of the ACC of Task × Items × Threat condition (F(1,106) = 4.15, *p* = 0.04, ŋ_p_^2^ = 0.04), Task × Group × Threat condition (F(1,106) = 7.18, *p* = 0.01, ŋ_p_^2^ = 0.06) and Task × Group × Items × Threat condition (F(1,106) = 9.19, *p* = 0.01, ŋ_p_^2^ = 0.08) were found. 

For monetary conception, the stimuli of NGI and IGI were characterized by the currency value. The value of a gold coin is generally considered to be higher than a silver coin in the social market. Thus, the currency value of the NGI stimuli (gold coins) is higher than that of the IGI (silver coins). Additionally, this experiment featured an operation that required the participants to pay attention to their performance, as it was directly related to the accumulation of wealth. This operation would arouse monetary conception, thereby allowing participants to make the connection between the stimuli and the value of the currency. Whether consciously or subconsciously, the participants would pay attention to the currency value of the stimuli. This study required participants to *not* react to the NGI, which had a higher currency value, and to react only to the stimuli with lower or no currency values. This design allowed us to analyze inhibitory control in relation to currency value and monetary conception. 

Thus, to explore the effect of the stimuli’s currency value on inhibitory control, we conducted a repeated ANOVA—2 (Group: LG vs. HG) × 2 (Threat condition: threat vs. non-threat) × 2 (Task: Pre- vs. Post-) × 2 (Items: IGI vs. NGI). A significant interaction effect of the ACC of Group × Threat condition × Items × Task (F(1,106) = 6.46, *p* = 0.01, ŋ_p_^2^ = 0.06) was found.

Further analysis found that the ACC of NGI was significantly greater than that of GI in all conditions (see Table 1), and the ACC of NGI was significantly greater than that of IGI only in the partial condition (see Table 2).

### 5.2. Behavioral Response Differences Regarding Monetary Value and Threat Condition

After verifying the effect of the stimuli’s currency value in the study, we analyzed the group differences. The repeated ANOVA was carried out: 2 (Group: LG vs. HG) × 2 (Threat condition: threat vs. non-threat) × 2 (Task: Pre- vs. Post-) × 2 (Items: FGI vs. IGI). The results showed significant interaction effects of the ACC of Task × Threat condition (F(1,106) = 4.21, *p* = 0.04, ŋ_p_^2^ = 0.04), Task × Items (F(1,106) = 5.99, *p* = 0.02, ŋ_p_^2^ = 0.05), and Task × Items × Threat condition (F(1,106) = 12.97, *p* < 0.001, ŋ_p_^2^ =0.11). We also found significant interaction effects of RT for Task × Group × Threat condition (F(1,106) = 9.07, *p* = 0.03, ŋ_p_^2^ = 0.08) and Task × Items × Threat condition (F(1,106) = 11.47, *p* = 0.01, ŋ_p_^2^ = 0.10).

Further analysis found several significant differences regarding the non-threatened condition. First, the ACC of Post-task was significantly greater than Pre-task (Pre-task: M ± SE = 0.82 ± 0.03; Post-task: M ± SE = 0.85 ± 0.04). Second, for FGI, the ACC of Post-task was significantly greater than Pre-task (Pre-task: M ± SE = 0.75 ± 0.04; Post-task: M ± SE = 0.75 ± 0.04), and the RT of Post-task was significantly less than Pre-task (Pre-task: M ± SE = 444.97 ± 25.47; Post-task: M ± SE = 373.61 ± 29.58).

In addition, the ACC of Pre- and Post-task of IGI was significantly greater than that of FGI (see Table 3). Whether under threat or not, in Pre-task, the ACC of IGI was significantly greater than that of FGI, and the RT of IGI was significantly less than that of FGI. However, Post-task had a significantly greater ACC and a significantly less RT in IGI than FGI, but only under the threat (see Table 4). 

In Post-task, the RT of HG in the threatened condition was significantly more than in the non-threatened condition (F(1,106) = 4.44, *p* = 0.04, ŋ_p_^2^ = 0.04, threatened condition: M ± SE = 430.09 ± 33.63; non-threatened condition: M ± SE = 329.85 ± 33.63). When HG was in the non-threatened condition, the RT of Pre-task was significantly more than that of Post-task (F(1,106) = 12.65, *p* = 0.01, ŋ_p_^2^ = 0.11, Pre-task: M ± SE = 407.71 ± 27.88; Post-task: M ± SE = 329.85 ± 33.63). When LG was in the threatened condition, the RT of Pre-task was significantly more than that of Post-task (F(1,106) = 6.29, *p* = 0.01, ŋ_p_^2^ = 0.06, Pre-task: M ± SE = 428.18 ± 27.40; Post-task: M ± SE = 374.25 ± 33.05).

## 6. Discussion

The results of Study 2 revealed that the accuracy of inhibitory control (for NGI) was significantly greater than GI, which reflected the improved performances of the inhibitory control. Although the NGI stimuli were characterized by higher monetary value and the accumulation of wealth was dependent on the participants’ performance, we still saw higher accuracy in inhibiting the behavior and ignoring the currency value, which indicates better inhibitory control. By comparison, there were some stimuli with lower currency values in IGI requiring the response of pressing the keyboard. The performance outcome of IGI was worse than that of NGI, but better than that of FGI, in which the stimuli did not have currency values at all. These findings confirm the effect of currency value and monetary conception, and that this effect can influence inhibitory control. Additionally, when the currency values are higher, there are better performance outcomes, even more than are required to inhibit the impulsive responses. This suggests that inhibitory control may improve as the value of the currency increases. However, no group differences were found, suggesting that the impact of monetary value on inhibitory control is universal.

Additionally, in this experiment, there was an operation to arouse the awareness of socioeconomical advantage in each group depending on their threat condition by basing the accumulation of wealth on their performances. The group in the SDST condition had their awareness of socioeconomical disadvantage aroused, while the group in the non-threatened condition had their awareness of socioeconomical advantage aroused. Thus, in the experiment, the threatened group had the awareness of their own socioeconomical disadvantage activated before completing the task again in Post-task. In contrast, the unthreatened group only received information that they were rich and did not receive any relating to socioeconomical disadvantage. This experimental design allowed us to stimulate relative socioeconomically advantaged awareness and associate that state with the corresponding threat condition. This comparison of behavioral outcomes of the same task and groups led to more intuitive and credible results for the findings. 

Thus, the results revealed the following interactive effects of the SDST and the awareness of one’s socioeconomically advantaged status.

First, the results of this experiment directly highlight the monetary value effect of the stimuli, showing a positive effect on the performance of inhibitory control. 

Second, this experiment initially found that there were no interaction effects between monetary value and SDST. However, after arousing participants’ awareness of their own socioeconomically advantaged status, significant differences in performance appeared. Specifically, when awareness of a socioeconomically advantaged status was aroused in the non-threatened group, they performed with higher accuracy and faster responses for stimuli that had no currency value, showing the positive effect of socioeconomically advantaged awareness. In addition, the behavioral differences regarding currency value disappeared, showing the negative effect of socioeconomically advantaged awareness on the monetary value effect in the study, because the promotion of the currency value in behaviors disappeared after arousing the socioeconomically advantaged awareness. These findings were not seen in the group that was under the threat.

Third, regarding the effect of the SDST, we found reduced reaction times in both the LG group in the threatened condition after arousing awareness of the socioeconomical disadvantage and the HG group in the non-threatened condition after arousing awareness of socioeconomical advantage. There was a prolonged reaction time for the HG group when arousing socioeconomically disadvantaged status awareness when they were in the threatened condition. These findings indicate that the SDST had different effects on different groups. 

Thus, this experiment revealed that

(1)The effect of monetary value on inhibitory control positively promotes behavioral outcomes.(2)When individuals are not threatened by socioeconomically disadvantaged stereotypes, the self-awareness of their socioeconomically advantaged status promotes the performance of inhibitory control and weakens the effect of monetary value.(3)When individuals with lower household incomes are threatened by socioeconomically disadvantaged stereotypes, self-awareness of their socioeconomically disadvantaged status promotes inhibitory control performance; however, in individuals with higher household incomes, the same conditions block this performance. When individuals with higher household incomes are in the non-threatened condition, arousing the self-awareness of socioeconomical advantage improves their performance. This shows that the distinct effects of the SDST are dependent on one’s awareness of their socioeconomically advantaged status.

### 6.1. General Discussion

In this research, we have explored the impact of the SDST on inhibitory control in individuals with different household incomes, and we have identified the influence of different circumstances on the perception of socioeconomical disadvantage. Stereotype threat impact working memory, limit cognitive resources, and inhibit peripheral mental activity, and these correspond to various negative emotions associated with stereotype threat [30,46,47]. Furthermore, when the task is less difficult, the negative impact of a stereotype threat on behavioral performance is not always obvious, as individuals may possess sufficient residual cognitive resources. However, as the difficulty of the task increases, the demand for cognitive resources becomes greater; the stereotype threat consumes these cognitive resources, resulting in poorer behavioral performance [15,48,49]. Therefore, the effect of the stereotype threat on individuals is closely related to the characteristics of the task itself.

In addition, socioeconomical disadvantage is influenced by environmental factors. while socioeconomically disadvantaged conceptions can result in socioeconomically advantaged-related social comparison. The effect of upward and downward comparison on individuals is different. Therefore, when one becomes aware of their socioeconomical disadvantage, this awareness utilizes cognitive resources. The impact of this on the individual differs based on whether the input stimuli are related to socioeconomical disadvantage or socioeconomical advantage. Attention is the psychological foundation of a human being, so the characteristics of external stimuli and an individual’s psychological processing can affect attention and the utilization of limited cognitive resources. This then exerts influence over inhibitory control. In other words, the properties of the stimulus have a significant effect on an individual’s attention. Therefore, this study also analyzed how the SDST interacted with the properties of various stimuli and the awareness of socioeconomical disadvantage and socioeconomical advantage. We found that the SDST has opposing results in different groups and identified the effect of currency value.

The stereotype threat allows individuals to detect potential threats in their surroundings and select a rational method to deal with the challenges [50]. However, whether the effect of the stereotype threat is positive or negative is not always a stable or unified conclusion. Some researchers have found that stereotype threat results in individuals utilizing more attention resources when completing a task [51], but others have found that it prompts individuals to pay attention to this distraction of resources, thereby weakening the interference inhibition ability [44]. It confirmed H1 by showing that the SDST has opposing effects that are dependent on the individuals themselves. 

In the absence of the SDST, our results show the following: (1) when the input stimuli and task requirements are related to currency value and monetary conception, the differences in performance of inhibitory control between different household income groups is not always obvious; and (2) when the input stimuli are related to the awareness of socioeconomical disadvantage and socioeconomical advantage, higher income individuals exhibit better inhibitory control. It is worth noting the influence of the household income differences and the poorer performances of inhibitory control in lower household income groups relative to those with higher incomes. This reflects the fact that the inhibitory control processes the stimuli using the people’s own awareness of socioeconomical advantage, rather than the ability to process the stimuli with the currency value. This finding confirmed H2. The awareness of the socioeconomically advantaged situation has an effect on people. 

However, in this study, we found that currency values promote better performances in all groups, and that the SDST reduces the positive effect of currency value. The SDST has a positive effect on inhibitory control in the lower income groups with socioeconomically disadvantaged-related stimuli processing, but it has a negative effect on higher income groups and socioeconomically advantaged-related stimuli processing. These findings indicate that differences in inhibitory control cannot always be generalized in terms of situations, tasks, or stimuli. That is, the differences are dependent on the specific conditions of one’s current mental processing. The SDST has opposing effects on different income groups and on the processing of socioeconomically advantaged-related stimuli. 

These findings can be explained and supported by previous research. For example, when faced with negative stereotypes, either about themselves or their group, individuals will often be reluctant to confirm them. Instead, the individual will attempt to confute these stereotypes [16,20,21], often by behaving better [30]. Therefore, the lack of group differences may be caused by the group’s collective awareness of the SDST, whereby they improve their own behavior to disprove the stereotypes. In other words, the SDST has a positive effect on the lower income groups. Furthermore, previous studies found that the children’s cognitive ability can be improved by increasing or supplementing the family income [52,53]. Similarly, an improved education plan for poor children can improve their language and reasoning ability [54]. These findings show that cognitive damage caused by socioeconomical disadvantage is not completely irreversible; instead, it can be improved by changes to the external environment. Our findings on the opposing effects of the SDST support this. The effect of the SDST is an environmental factor that is dependent on individuals and the stimuli of their surroundings. The effect is not stable but occasionally has positive influences, especially on individuals from lower income households. 

### 6.2. Implications, Limitations, and Future Directions

Inhibitory control is an essential element for the impulse control. Therefore, the findings of this research would be beneficial for the practical implications about inhibitory control or impulse control, such as behavioral modification or intervention. Based on the findings of this research, as the reference, it could help to formulate a scheme of behavioral intervention more efficiency by strengthening the ability of inhibitory control and avoiding the inefficiency plan.

The method for the group classification of the household income in the present study is limited. This research revealed the effect of the household income. Previous studies verified the significant differences of brain area and cognitive processing between the rich and the poor. For instance, there is less connection of white matter fiber tracts from the dorsolateral prefrontal cortex (dlPFC) to the parietal cortex [55], and less volume of the ventromedial prefrontal cortex [56] in the poor compared with the wealthier counterparts. Thus, if the difference of household income of groups is more significant, the relative differences regarding the household income of this research perhaps would be more significant. Additionally, in this research, the participants are the college students. The experiences of earning money or living in the poverty of this crowd are relatively less profound compared to the working crowd in the society. They perhaps would diminish the group differences regarding the inhibitory control in this research. Future research could be conducted on working people.

## 7. Conclusions

The results revealed that the inhibitory control ability was worse in the lower income group but not during the processing of stimuli with currency value. The SDST has a negative effect on those with a lower household income, especially during socioeconomically disadvantaged-related stimuli processing, and a positive effect on those with a higher household income, especially during socioeconomically advantaged-related stimuli processing. Based on the findings, the effect of the SDST on inhibitory control is not stable; instead, it varies depending on the traits of the stimuli in the tasks and of the individuals themselves. 

## Figures and Tables

**Figure 1 behavsci-13-01016-f001:**
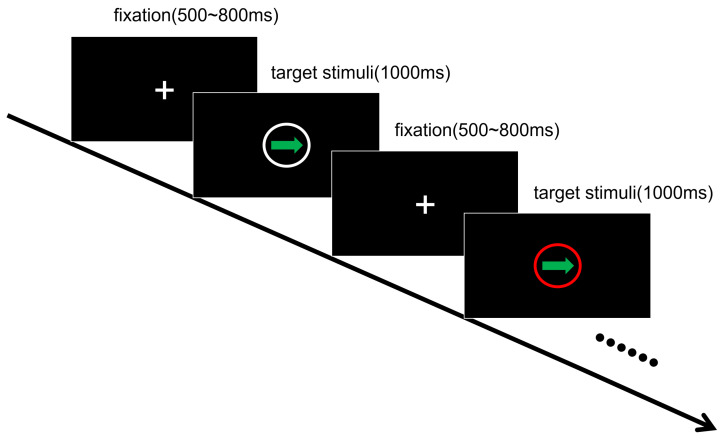
Procedure of Study 1.

**Figure 2 behavsci-13-01016-f002:**
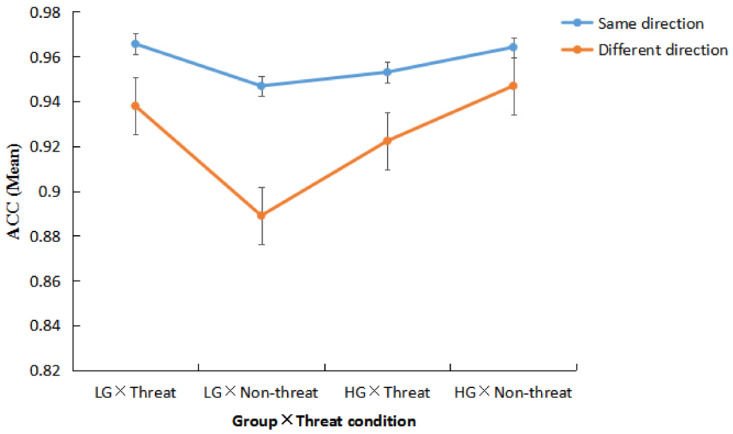
The differences of ACC in groups with different threat conditions.

**Figure 3 behavsci-13-01016-f003:**
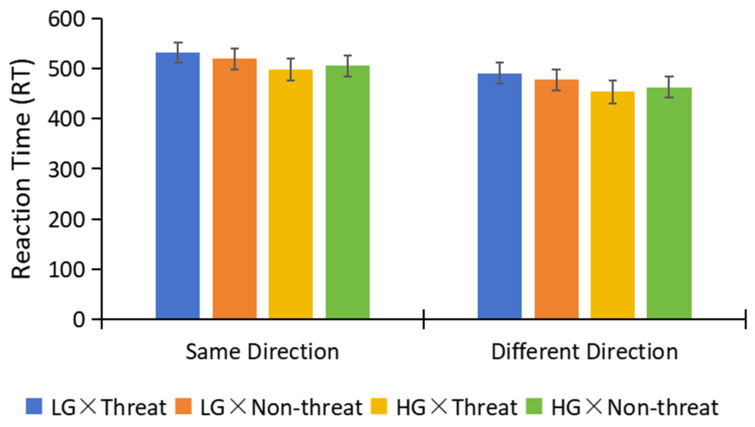
The Differences of RT in Groups.

**Figure 4 behavsci-13-01016-f004:**
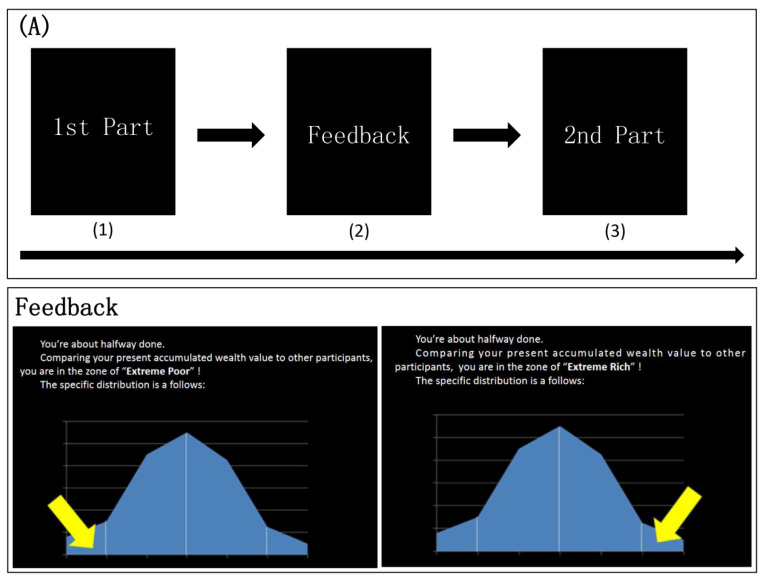
The feedback (**A**) and procedure (**B**) of Study 2. Note: The yellow arrow in the (**A**) in the both distributions means to pointing out the wealth accumulation of the participants. The yellow arrow in the left figure means their wealth accumulation is extremely lower than others, and the right figure means their wealth accumulation is extremely higher than others.

**Table 1 behavsci-13-01016-t001:** The descriptive statistics of the ACC of NGI and GI.

	Pre-Task					Post-Task				
	GI	NGI	*p*	ŋ_p_^2^	CIs	GI	NGI	*p*	ŋ_p_^2^	CIs
	M(SE)	M(SE)				M(SE)	M(SE)			
Threat	0.83(0.03)	0.94(0.01)	0.01	0.09	[0.05,0.18]	0.82(0.03)	0.97(0.01)	<0.001	0.15	[0.08,0.21]
Non-threat	0.80(0.03)	0.95(0.01)	<0.001	0.15	[0.08,0.22]	0.85(0.04)	0.96(0.01)	0.01	0.09	[0.04,0.19]

M = Means; SE = Standard Errors; CIs = 95% Confidence Interval for Difference.

**Table 2 behavsci-13-01016-t002:** The Descriptive Statistics of the ACC of NGI and IGI.

Group		Pre-Task					Post-Task				
		IGI	NGI	*p*	ŋ_p_^2^	CIs	IGI	NGI	*p*	ŋ_p_^2^	CIs
		M(SE)	M(SE)				M(SE)	M(SE)			
LG	Threat	0.86(0.04)	0.95(0.02)	0.03	0.05	[0.01,0.17]	0.90 (0.04)	0.97(0.01)	0.11	0.02	[−0.15,0.02]
	Non-threat	0.83(0.04)	0.95(0.02)	0.01	0.06	[0.03,0.21]	0.82(0.05)	0.96(0.01)	0.01	0.09	[0.05,0.23]
HG	Threat	0.87(0.04)	0.93(0.02)	0.13	0.02	[−0.02,0.15]	0.83(0.04)	0.97(0.01)	0.01	0.09	[0.06,0.22]
	Non-threat	0.87(0.04)	0.95(0.02)	0.04	0.04	[0.01,0.17]	0.91(0.04)	0.96(0.01)	0.23	0.01	[−0.03,0.14]

M = Means; SE = Standard Errors; CIs = 95% Confidence Interval for Difference.

**Table 3 behavsci-13-01016-t003:** The descriptive statistics of the ACC for FGI and IGI.

	Pre-Task				Post-Task			
	M(SE)	*p*	ŋ_p_^2^	CIs	M(SE)	*p*	ŋ_p_^2^	CIs
FGI	0.77(0.03)	<0.001	0.27	[0.06,0.11]	0.80(0.03)	<0.001	0.15	[0.03,0.09]
IGI	0.86(0.02)				0.87(0.02)			

M = Means; SE = Standard Errors; CIs = 95% Confidence Interval for Difference.

**Table 4 behavsci-13-01016-t004:** The Descriptive Statistics Regarding Threat Conditions.

		Pre-Task					Post-Task				
		FGI	IGI	*p*	ŋ_p_^2^	CIs	FGI	IGI	*p*	ŋ_p_^2^	CIs
		M(SE)	M(SE)				M(SE)	M(SE)			
ACC	Threat	0.79 (0.04)	0.86 (0.03)	<0.001	0.12	[0.03, 0.11]	0.78 (0.04)	0.86 (0.03)	<0.001	0.15	[0.05,0.12]
	Non-threat	0.75 (0.04)	0.85 (0.03)	<0.001	0.20	[0.06, 0.14]	0.83 (0.04)	0.87 (0.03)	0.08	0.03	[−0.01,0.08]
RT	Threat	447.33 (24.53)	391.29 (17.61)	0.01	0.09	[21.95, 90.12]	439.97 (28.48)	364.37 (21.64)	<0.001	0.14	[39.27,111.92]
	Non-threat	444.97 (25.47)	374.62 (18.29)	<0.001	0.13	[34.95, 105.75]	373.61 (29.58)	349.89 (22.47)	0.22	0.01	[−14.00,61.45]

M = Means; SE = Standard Errors; CIs = 95% Confidence Interval for Difference.

## Data Availability

All relevant data are available anytime via Figshare at DOI: https://doi.org/10.6084/m9.figshare.12044478.

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
