# Peer review of "The Effect of Socioeconomically Disadvantaged Stereotype Threat on Inhibitory Control in Individuals with Different Household Incomes"

_behavsci, 2023, doi:10.3390/bs13121016_

Round 1

Reviewer 1 Report

Comments and Suggestions for Authors

Thank you for allowing me to review your manuscript. I have addressed my edits below.

In the introduction, please introduce the acronym PST before using it consistently.

Also, please communicate the so what of your study in the introduction. You do a good job of defining terms, but I am unsure of the motivation to pursue such a study.

There is not a purpose statement. This study would benefit from one.

Is there a theoretical underpinning to this study?

I would also encourage a section titled “Hypothesis Development” or something similar to move the reader along better. Same with a “Methods” section.

What were the stimuli used?

What constitutes a lower income and higher income family?

How were the participants recruited?

What was the demographic breakdown of your participants?

Why was this procedure adopted for Study 1 and Study 2? Can you please provide some background or foundation?

Overall, more is needed in the methodology sections of these studies.

Overall, what are the real-world implications for this study? It is interesting, but why conduct this? This needs to be addressed more in the conclusion and an implication section.

Please format your in-text citations to the journal requirements.

Please update literature with more recent citations.

Comments on the Quality of English Language

Please conduct a minor edit to the manuscript.

Author Response

Thanks for the suggestion of the reviewer, and the point-to-point response to the comments were in the uploaded document. 

Reviewer 2 Report

Comments and Suggestions for Authors

This manuscript investigates the effects of poverty stereotype threat on decision making abilities of young individuals. The authors have conceived 2 interesting studies to test their hypothesis, however some concerns remain.

 Major concerns –  

 1.      The concepts of general stereotype threat and poverty stereotype threat can be better defined in the introduction. The authors mention ‘…the typical effect of stereotype threat’ in second paragraph of introduction. However, since these concepts are crucial for the study, it is better to state these definitions and the rationale of the study clearly.

 2.      Some sentences in the introduction like ‘…When all participants were from lower-income families, the participants who were informed that they were taking a intelligence test were worse than those who were not informed (Spencer & Castano, 2007, Har-rison et al., 2006)…’ are identical to a previous publication by the authors Brain and Behavior. 2020;10:e01770.

 3.      Were there any specific inclusion criteria for the participants? Were any subjects excluded later from the study? The authors mention, ‘…Fifty-three participants were from families with lower household income group (LG). The remaining participants were belonging to the higher household income group (HG; N=56)…’ What was the criteria for determining low-income group vs high income group?

 4.      The authors mention in the Participants section ‘…Twenty-eight LG participants and 28 HG participants were under the PST, and the rest of the participants were not (LG N=25, HG N=28)...’. How was threat condition assessed and determined for subjects of low- and high-income groups? It is important to clearly mention in methods section the criteria used to determine threat condition as it is key to the study. Were subjects randomly assigned to threat conditions or were any other methods applied?

 5.      A pictorial representation of the study design for Study 1 and Study 2 would be good.

 6.      In the results of Study 1 the authors mention there were no significant effects on reaction times. It would be better to plot the data as a figure perhaps.

 7. In Study 2, the threat condition participants received feedback on how less their wealth was in compared to others and participants of the non-threat condition received feedback on how significantly higher their wealth was compared to others. Was any data collected for the same participants after swapping the feedback to test if individuals in threat condition group respond better in GI-IGI-FGI task after receiving feedback on how significantly higher their wealth was compared to others. Similar concern for non-threat group to check if their performance in GI-IGI-FGI task worsens after receiving feedback on how less their wealth was in compared to others.

Comments on the Quality of English Language

The overall English language can be improved throughout in the manuscript.

Author Response

(The authors gave the same response as above.)

Reviewer 3 Report

Comments and Suggestions for Authors

Thank you for your submission.

This is an intriguing study.

While I don't have significant concerns, there are a few parts that could be addressed to enhance scientific communication and improve readability:

1. It might be beneficial for the authors to use a less stigmatising term than 'poverty.' Language can be tempered to avoid misinterpretation or potential ‘offensive interpretation’ by readers. Additionally, the connection between 'stereotype threat,' inhibitory control, and various household factors could be clearer;

2. Could you elaborate on the rationale behind choosing inhibitory control? Providing a link or explanation about why this variable is crucial could better engage readers, especially those not familiar with this specificity;

3. I'm curious about the involvement of problem-solving, especially considering interesting potential findings from previous studies using electrophysiology;

4. Please enlist a native English speaking colleague to edit the ms for language refinement;

5. Ensure there are no blank spaces and maintain coherence between paragraphs for better flow;

6. If possible, adopting a more formal language in certain parts of the manuscript could be beneficial;

7. Consider maintaining consistency in sentence length or paragraph structure throughout the manuscript;

8. Do you believe it's essential to include H1 and similar elements for readers' understanding?;

9. Provide detailed eligibility criteria and reasons for exclusion, as well as how missing data were handled;

10. Consider using an illustration or flowchart to represent the study's workflow for better comprehension;

11. While the sample size seems adequate, consider modifying the formula to conduct more robust analyses (i.e. changing the f) and please include attrition rate;

12. Expand the stat: section to include relevant information regarding tests, analyses, effects, and CIs;

13. Consider utilising Bayesian, and if interested in, software that simplifies these calculations are free and quick to use. Also, clarify why not use LMM instead of inflating the model with multiple analyses. For multiple dependent variables, consider MANOVA;

14. Running several similar tests may bias the model and inflate the data, so please be cautious;

15. HQ illustrations are recommended. There is software available to assist in creating these. A scatterplot accompanied by boxes could be particularly useful;

16. Improve the clarity of schematics by potentially altering the font or extending the file to enhance better interpretation;

17. Consider using Omega regardless of eta, as partial effects may not be entirely reliable, especially without considering dispersion using CIs;

18. Provide clear captions for the abbreviations used in Tables (e.g. M, SE);

19. Please check your refs.;

Though these suggestions may seem extensive, and even annoying, they are intended to be helpful. 

Wishing you success with the study

Comments on the Quality of English Language

Please carefully check grammar and punctuation 

Author Response

(The authors gave the same response as above.)

Round 2

Reviewer 1 Report

Comments and Suggestions for Authors

Thank you for reviewing and making changes to your manuscript. Please see the follow-up suggestions.

I still do not see the purpose statement. Please add to the Introduction of the paper. 

You state "Please believe the studies were conducted on the scientific underpinning." This needs to be added to the manuscript to provide a theoretical background for your study. Your readers will not assume this.

Thank you for adding the stimuli and an implication section.

Author Response

Thank you very much for taking the time to review this manuscript. Please find the detailed responses below and the corresponding revisions/corrections highlighted in blue ink in the re-submitted files.

Reviewer 2 Report

Comments and Suggestions for Authors

The authors have revised the manuscript extensively and addressed most of my concerns. However, 1 major concern continues to persist.

Major concerns – 

1.      The authors mention – “…Furthermore, the participants were assigned to the threatened condition randomly by the traditionally threatened method. The survey results were checked whether the participants were under the threatened condition…”. It is not still clear what is the traditionally threatened method and which survey results were used. If the authors can include few more sentences regarding this in the methods section that would be fantastic please.

Comments on the Quality of English Language

The language editing of the manuscript has been improved considerably from previous submission. However, there is scope for further improvement in the introduction and methods section.

Author Response

(The authors gave the same response as above.)

Reviewer 3 Report

Comments and Suggestions for Authors

Thank you for your careful and thoughtful edits, commended for the endeavour in addressing the concerns.

Nevertheless, there are concerns important not addressed:

1. Consider changing the sample formulae. Changing the ‘f’ will ensure a more robust analysis plus using the attrition rate is just the basics from best practices. This won’t change that much and won’t make authors needing changing sample;

2. The use of scatterplots are essential. I’m a little bit surprised the authors did not consider it. The use of scatter will present the readers dispersion of individual data. The use of box plus scatter / dots is the best or one of the best approaches to show how your data behaved - the use of more illustrative presentations would benefit the ms, but box / scatter is simplistic and yet useful 

Author Response

(The authors gave the same response as above.)
